# Lateral Structured Phototransistor Based on Mesoscopic Graphene/Perovskite Heterojunctions

**DOI:** 10.3390/nano11030641

**Published:** 2021-03-05

**Authors:** Dahua Zhou, Leyong Yu, Peng Zhu, Hongquan Zhao, Shuanglong Feng, Jun Shen

**Affiliations:** Institute of Green and Intelligent Technology, Chinese Academy of Sciences, Chongqing 400714, China; lyyu@cigit.ac.cn (L.Y.); zhupeng@cigit.ac.cn (P.Z.); hqzhao@cigit.ac.cn (H.Z.); fengshuanglong@cigit.ac.cn (S.F.); shenjun@cigit.ac.cn (J.S.)

**Keywords:** graphene nanowalls, perovskites crystal, phototransistor, heterojunctions

## Abstract

Due to their outstanding optical properties and superior charge carrier mobilities, organometal halide perovskites have been widely investigated in photodetection and solar cell areas. In perovskites photodetection devices, their high optical absorption and excellent quantum efficiency contribute to the responsivity, even the specific detectivity. In this work, we developed a lateral phototransistor based on mesoscopic graphene/perovskite heterojunctions. Graphene nanowall shows a porous structure, and the spaces between graphene nanowall are much appropriated for perovskite crystalline to mount in. Hot carriers are excited in perovskite, which is followed by the holes’ transfer to the graphene layer through the interfacial efficiently. Therefore, graphene plays the role of holes’ collecting material and carriers’ transporting channel. This charge transfer process is also verified by the luminescence spectra. We used the hybrid film to build phototransistor, which performed a high responsivity and specific detectivity of 2.0 × 10^3^ A/W and 7.2 × 10^10^ Jones, respectively. To understand the photoconductive mechanism, the perovskite’s passivation and the graphene photogating effect are proposed to contribute to the device’s performance. This study provides new routes for the application of perovskite film in photodetection.

## 1. Introduction

Graphene, nicknamed “miracle material”, a typical two-dimensional film was well explored experimentally, as well as theoretically and computationally, which provided an attractive platform for broadband visible-infrared photodetection [1,2,3]. Many efforts were devoted to designing graphene-based photodetector; however, such two-dimensional material presented a weak absorption and a low responsivity. Considering the graphene’s ultrahigh carrier mobility, to enhance the detector’s responsivity, it is easy to come up with the idea to hybrid it with a material which owns a high absorption, such as the most used semiconductor silicon, quantum dots as CdS, PbS, and organic–inorganic halide perovskites, etc. [4,5,6,7,8].

Due to the properties of high absorption coefficient, long charge carrier diffusion length, and excellent mobility, the organic–inorganic halide perovskites are extensively studied, which can be described by the formula CH_3_NH_3_PbX_3_ (X = Cl^−^/Br^−^/I^−^) [9,10]. Besides high-efficiency solar cells, perovskites showed great application prospects in light-emitting diodes, nanolasers, and photodetectors [11,12,13,14]. In fact, the past years have witnessed the rapid progress of various perovskite-based photodetectors made from one-dimensional (1D) nanowire, 2D ultrathin film, and 3D perovskite crystals. For instance, scientists reported that a oleic acid–soaking passivated CH_3_NH_3_PbI_3_ nanowire photodetector possessed a high detectivity and polarization sensitivity characteristics [15]. Furthermore, with an ultrathin single-crystalline perovskite film, Mu Wang’s team successfully synthesized a high-performance flexible photodetector with prevailing bending reliability [16]. Meanwhile, perovskite-based photodetectors are always fabricated as vertical or lateral structures. The former one is composed layer-by-layer, which is in favor of carrier separation and results in a low noise and high detectivity [17]. Compared with that, however, the lateral structure photodetectors or solar cells are simple enough and can be much more easily fabricated [18,19].

By combining the graphene’s high carrier mobility with the perovskites’ good light-harvesting capability, high-performance graphene/perovskite photoelectric devices have been widely demonstrated [20,21,22,23,24,25,26]. Indeed, Wang and co-workers obtained an ultrahigh photoresponsivity, i.e., 6.0 × 10^5^ A/W, with a graphene-perovskite phototransistor [20]. Other works, such as CH_3_NH_3_PbI_3_ nanowire/graphene, gold plasmonic-enhanced graphene/perovskite photodetectors, and even fully inkjet-printed graphene/perovskite/graphene photodetector have been investigated [21,22]. A key mechanism in these hybrid structures is that photon-induced holes transfer from perovskite to graphene, leaving the excited electrons at conduction band without decaying, thus enhancing the photo-excited carrier’s lifetime, and further increasing the photoresponsivity. Such effects closely connected with the charge transfer process in the interfacial, referred to as the photogating effect or photodoping effect [8,23]. 

Graphene nanowalls’ (GNWs) structure in a nanoframework is vertical oriented and possessed many unique properties, such as mechanical stability, high in-pane conductivity, and high surface-to-volume ratio [27,28]. Considering the graphene nanowalls’ porous structure and the perovskite’s polycrystalline property, it is much effective for photo-excited holes to transfer at the perovskite/graphene interface. Based on this concept, here we devised a graphene/perovskite hybrid phototransistor in this work. Due to their energy band alignment gap, a heterojunction formed at the interface. The distance between the graphene plate is comparable with the electron–hole diffusion length in perovskite crystalline, thus, more excited carriers could be connected by the electrodes but not recombined [9]. This lateral structure phototransistor not only promotes the application of graphene and perovskite in photodetection, but also indicates enormous economic advantages in solar cell application.

## 2. Experimental Section

### 2.1. Graphene Nanowalls Fabrication

Graphene nanowalls were fabricated on a Si/SiO_2_ (300 nm) substrate, using the radio-frequency plasma-enhanced chemical vapor deposition (RF-PECVD) method. Before growth, the substrate was cleaned in acetone, ethanol, and distilled water, respectively. A mixture gas of methane and hydrogen (6:4) was introduced as the reactant. The growth temperature was about 800 °C, with a chamber pressure of 50 Pa. During growth, the radio frequency power was set as 200 W, maintaining for 45 min for graphene growth.

### 2.2. Perovskite Preparation

The CH_3_NH_3_PbI_x_Cl_3-x_ perovskite was synthesized according to the method reported previously [29]. Raw materials of 1.26 M PbI_2_ (581 mg, 99.999%, Sigma-Aldrich, Saint Louis, MO, USA), 0.14 M PbCl_2_ (39 mg, 99.999%, Sigma-Aldrich), and 1.3 M CH_3_NH_3_I (209 mg, Dyesol, Queanbeyan, Australia) were mixed in co-solvent, at 60 °C, with stirring, that comprised of dimethyl sulfoxide (DMSO, anhydrous, ≥99.9%, Sigma-Aldrich) and γ-butyrolactone (γ-GBL, 99%, J&K, Beijing, China), with a ratio of 3:7. After coating, chlorobenzene was quickly added to the substrate, to induce crystallization, and the film was annealed at 100 °C, for 10 min. The synthesis, coating, and annealing process of perovskite were finished in a glove box.

### 2.3. Device Fabrication

With a metal mask, the grown graphene nanowalls were etched into a 2 × 2 mm^2^ square by an oxygen plasma. Then, electrodes were deposited, using magnetron sputtering technique. At last, with another metal mask, perovskites were coated on the graphene nanowalls. Finally, the device was packaged in N_2_ atmosphere, as a prototype transistor.

### 2.4. Characterization and Photodetection

At first, Raman spectra of GNWs were obtained by using Renishaw inVia Reflex (Renishaw, Gloucestershire, UK) with a 532 nm laser. Scanning electron microscopy (SEM) and Transmission Electron Microscope (TEM) images were obtained by using JSM-7800F (JEOL, Tokyo, Japan) and JEM-2100F (JEOL, Tokyo, Japan), respectively. The crystal structure of the film was investigated by XRD measurements, with a PANalytical-X’Pert Powder (PANalytical, Amsterdam, Holland) diffractometer equipped with Cu-K radiation (0.154056 nm wavelength). Excited at 532 nm laser, the luminescence spectra of perovskites samples were determined on a FLS 980 spectrometer (Edinburgh Instruments, Edinburgh, UK) detected by red sensitive PMT. The decay curves at 770 nm were measured by using a 472.0 nm EPL-470 laser with a pulse width of 95.9 ps. In addition, the UV−visible excitation spectra were recorded on an Edinburgh FLS980 spectrometer equipped with a 450 W xenon lamp. To evaluate its photo-detective property, a solar simulator (XES-50S1-EI) and several laser sources (450, 532 and 633 nm) were used as the irradiation light, and a Keithley 2450 and Keithley 4200 (Tektronix, Shanghai, China) were used to record the electrical results, respectively. At last, we measured, directly, the noise spectral density of the dark current by using an FFT spectrum analyzer (MA-007, Measureact Tech., Shenzhen, China). All the measurements were performed at room temperature.

## 3. Results and Discussion

### 3.1. Characterization of Graphene/Perovskite Film

At first, the GNWs were grown by the RF-PECVD technique. Figure 1a shows a typical SEM image of GNWs grown on the Si/SiO_2_ substrate, which presents a porous structure and a large number of graphene edges. The graphene edges can also be testified by the Raman spectra in Figure 1b. Since the D peak (~1350 cm^−1^) requires a defect for its activation and the G peak (1580 cm^−1^) corresponds to the doubly degenerate zone center E_2g_ mode, we can see that, the longer the growth time of GNWs used, the larger the intensity ratio of the D to G (I_D_/I_G_), which means more graphene edge defects and a higher degree of disorder shaped [30,31]. However, from the increasing intensity of the D peak and the increasing full width at half maximum (FWHM) of the 2D peak, it could be inferred that a longer growth time leads graphene nanosheets to nanocrystalline graphite. Therefore, an appropriate growth time should be chosen to synthesize the graphene nanowalls. The TEM image presented a folding surface of the GNWs, and its thickness is about 10 layers, as shown in Figure 1c,d, respectively. It is expected that the porous structure supplies enough space for perovskite crystals to fill in, and the graphene nanowalls have a large surface connected with the perovskite crystalline grain boundaries.

Briefly speaking, by the step of GNWs etching, electrodes deposition, and perovskite spin-coating, a graphene/perovskite hybrid phototransistor was finally fabricated. Figure 2 presents the device’s schematic diagram and its structural characteristics. The cross-sectional scanning electron microscopy image showed that the perovskite microcrystalline filled in the graphene walls, and the hybrid film had a thickness of 0.8 μm (Figure 2c). The top-view SEM presented that not all the graphene films were covered by the perovskite crystal and the naked graphene located at the perovskite crystalline boundary (Figure 2d). Most perovskite crystalline grains have a size of less than 0.5 μm. In addition, after the thermal annealing process, for the perovskite grain, one can see that its boundary areas present relatively brighter than its center part. The bright areas are considered as PbI_2_, which is less conductive and accumulates more charges. This phenomenon indicated that a PbI_2_ phase was formed, which is referred to as the passivation effect [32]. Moreover, the PbI_2_ phase can be confirmed in the XRD pattern (Figure 2e). One can see that, for a same annealing time, the PbI_2_ peak in graphene/perovskite showed less intensity than that of pristine perovskite, possibly indicating a low PbI_2_ component formed in the hybrid film. In some sense, however, these PbI_2_ impurities can enhance the device’s photoelectric performances [33,34,35,36,37].

### 3.2. Optical Properties

The CH_3_NH_3_PbI_x_Cl_3-x_ perovskite has an absorption band ranging from 330 to 780 nm (Figure 3a) and a strong luminescence spectrum peaking at about 770 nm. In Figure 3b, one can see that, when we combined perovskite with graphene monolayer, the luminescence quenched, which has also been observed before [8,20]. Furthermore, for graphene/perovskite structure, the luminescence quenching presented much more intensity. It may be induced by a higher charge transfer probability and a lower luminescence recombination ratio. Thus, we can say, this is the direct consequence of the charge-carrier extraction, which happened across the graphene/perovskite interfaces. In addition, the excitation spectra of perovskite and its hybrid films present big differences, as shown in Figure 3c. We can find that, in the range of 300~450 nm bands, perovskites combined with graphene monolayer or nanowalls have a similar curve and a higher intensity than that of the pristine perovskite. However, when the excitation wavelength is longer than 520 nm, the luminescence intensity of pristine perovskite film rises sharply. In addition, no matter the pristine perovskite or the hybrid film, the luminesce lifetime at 770 nm has two branches, as shown in Figure 3d. The fast decay component, τ_1_, has been suggested to come from bimolecular recombination, and the longer one, τ_2_, was attributed to free carriers’ recombination [32]. In Table 1, one can find that both the fast and long components of the perovskite hybrid with graphene monolayer or nanowalls exhibit the same scale of lifetime as the previous results.

### 3.3. Performances of Graphene/Perovskite Photodetection

At first, the device’s photoelectrical response was tested under a solar simulator, as shown in Figure 4. Under different solar intensities, namely 0.5, 0.75 and 1 sun (1000 W/m^2^), the transistor presented an obvious photo-responsive property. With a switch-on and switch-off process, the I–t curves presented good stability (Figure 4b). In addition, we studied the performance of the phototransistor upon monochromatic light. With a bias voltage of 1 V, Figure 5a shows the transistor’s photocurrent response under different light radiation, in which one can find such device presented an obvious photocurrent response in the red, green, and blue light. Next, we investigate its photo responsibility with different incident light intensities. In the I–t curves of Figure 5b, by increasing the light power, higher photocurrent was obtained. Moreover, a typical response time was about 50 and 290 milliseconds for rise and fall time, respectively, which was much faster than that of the single layer graphene/perovskite photodetector [8,20,21]. Furthermore, the photocurrents and the corresponding responsivity of the graphene/perovskite photodetector were calculated by the following equations [39]:(1)$$I_{ph}=I_{illuminated}-I_{dark}$$
(2)$$R_{I}=\frac{I_{ph}}{P}$$
(3)$$D^{*}=R\frac{\sqrt{A*\Delta f}}{i_{n}}$$

The maximum responsivity of was 2.0 × 10^3^ A/W measured with an incident power of 40 μW of 635 nm laser. Considering the noise spectral density of the dark current, *i_n_* = 5.5 × 10^−9^ A/Hz^−1/2^ at the frequency of 1 Hz, the specific detectivity was calculated to be 7.2 × 10^10^ Jones. As listed in Table 2, compared with the responsivity reported in the references, such graphene/perovskite phototransistor performed a much higher responsivity and specific detectivity than that made of perovskite and single layer graphene film [8]. It could be ascribed to the efficient charge transfer process at the graphene wall’s interface. Another phenomenon that should be pointed out is that extremely low excitation power is helpful to get a higher responsivity in these optoelectronic experiments, especially for the nanoscale device; that is why high responsivity or specific detectivity was obtained in the perovskite detectors in References [8,20,21]. This phenomenon may be related to the complex process of carrier trapping and regeneration process, which is commonly observed in other graphene or perovskite photodetectors [24]. Moreover, it could also explain the decreasing photoresponsivity with increasing power in Figure 5d.

### 3.4. Operation Mechanism in the Graphene/Perovskite Phototransistor

For the highly performance hybrid photodetector based on graphene/perovskite film, a key theoretical hypothesis is that photo-excited holes in perovskites transfer to graphene, and therefore electrons in graphene transfer to a proximal perovskite layer (Figure 6). Compared with 2D graphene film, 3D graphene walls have a much larger contact interface with perovskite; therefore, it is expected that the charge transfer process should be more effective. After electron–hole pairs were exited in the perovskite, the holes were injected into the graphene, leaving electrons trapped in the perovskite islands. Due to this carrier separation in different material and the decreasing of carrier recombination, a hole’s lifetime is increased, thus enhancing the device’s responsibility. This current enhancement around the perovskite crystals was observed by the photocurrent mapping results, namely photogating effect [20].

Moreover, the passivation effect and photogating effect must be pointed out. Since the passivation process, PbI_2_ crystals separate from perovskite. These PbI_2_ crystal layers reduced the carrier recombination in the absorber superficies, as well as the grain boundary; therefore, they promoted the device performance [36]. In brief, the passivation effect in perovskite and the photogating effect when hybrid with graphene nanowalls both contribute to the device’s photodetection performance, by reducing the carrier recombination efficiently.

## 4. Conclusions

In summary, we have fabricated the first graphene/perovskite phototransistor. It is lateral structured and easy to be fabricated. By the spin-coating method, perovskites were infiltrated in the porous graphene nanowalls. In the device, graphene acted as the hole collecting and transfer layer, and mesoscopic heterojunctions formed in the graphene/perovskite interface. At room temperature, photoelectrical measurements show that it has a responsivity and specific detectivity of 2.0 × 10^3^ A/W and 7.2 × 10^10^ Jones, respectively. In addition, it achieved faster rise and fall times than that of the perovskite hybrid with single-layer graphene. Comparing with the with monolayer graphene hybrid with perovskite, this device based on graphene nanowalls showed a much high performance in photodetection. The hybrid phototransistor described here is expected to contribute to the development of lateral devices based on perovskite, such as the photodetection and solar cell applications.

## Figures and Tables

**Figure 1 nanomaterials-11-00641-f001:**
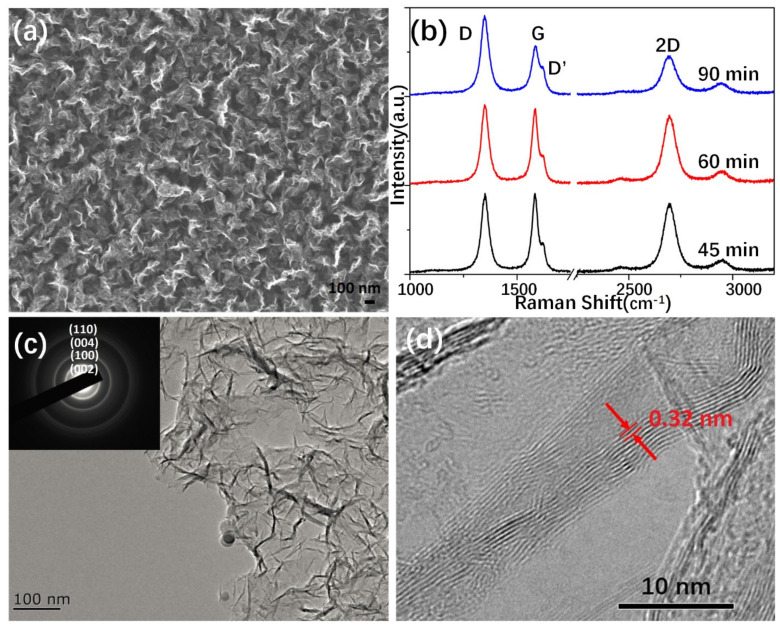
Characteristics of graphene nanowalls. (**a**) SEM image of the wrinkled graphene nanowalls grown on silicon. (**b**) Raman shift of the graphene grown in different time, laser: 532 nm. (**c**) TEM image and (**d**) HRTEM image of graphene. The inset in (**c**) is the selected area electron diffraction (SAED) pattern of the graphene nanosheets.

**Figure 2 nanomaterials-11-00641-f002:**
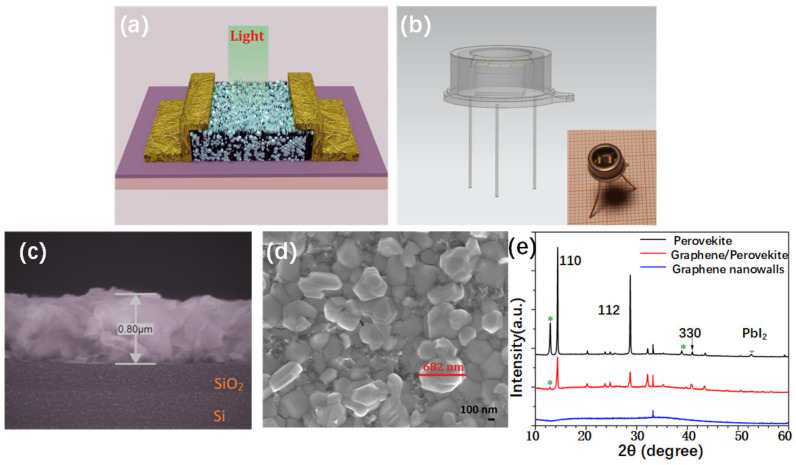
Schematic diagram of the device and characteristics of the graphene/perovskite hybrid photodetector. (**a**,**b**) Configuration of the lateral structured grapheme/perovskite phototransitor and the fabricated prototype device. (**c**) Cross-sectional image of the device. (**d**) Top-view SEM image of the graphene/perovskite hybrid film. (**e**) XRD patterns of the graphene nanowalls on silicon, graphene/perovskite film, and pure perovskite crystals;.

**Figure 3 nanomaterials-11-00641-f003:**
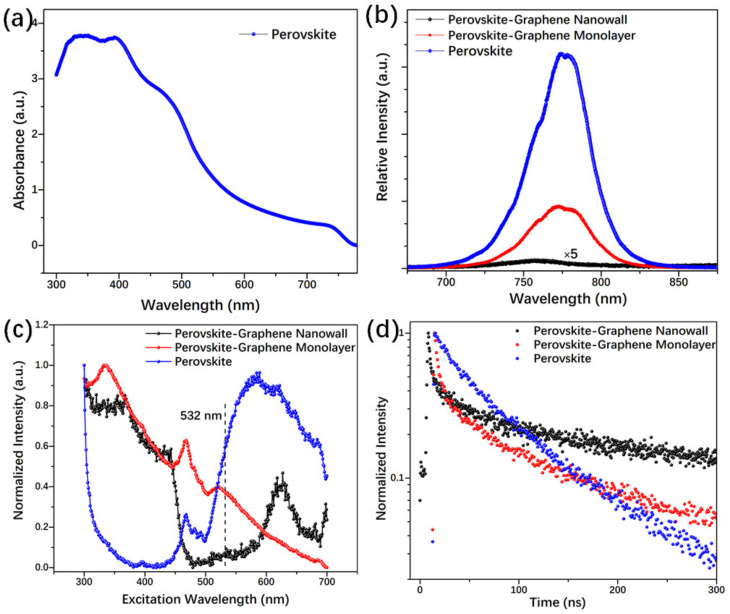
(**a**) UV–visible absorption spectra of perovskite. (**b**) Excited at 532 nm, luminescence spectra of pure perovskite, perovskite/graphene monolayer, and perovskite/graphene nanowalls. (**c**) Excitation spectra of various perovskite samples, λem = 770 nm. (**d**) Time-resolved luminesce decay curves of perovskite samples measured at 770 nm.

**Figure 4 nanomaterials-11-00641-f004:**
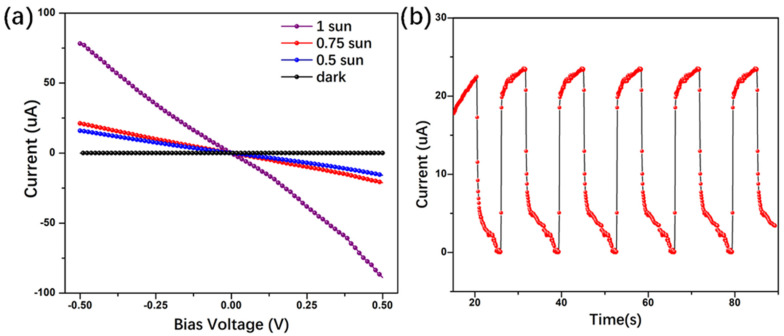
(**a**) Drain current–drain voltage plot irradiated by solar simulator, 1 sun = 1000 W/m^2^. (**b**) Photo-switching characteristics of the graphene/perovskite phototransitor under alternating dark and light illumination (1sun, 0.25V).

**Figure 5 nanomaterials-11-00641-f005:**
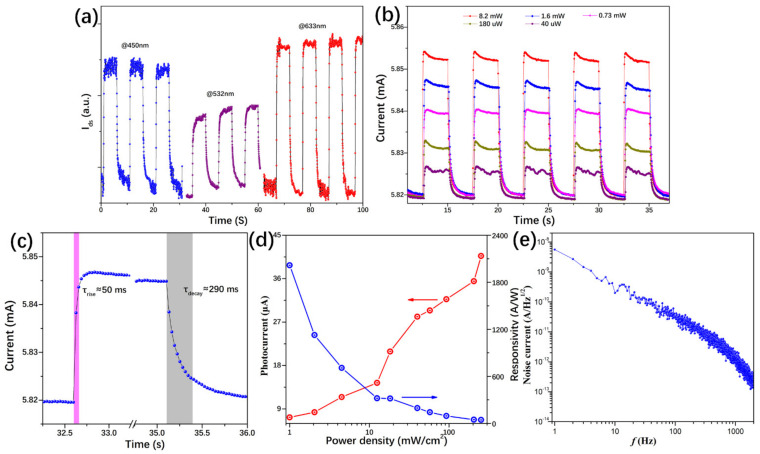
(**a**) With a 1 V voltage, the photocurrent response to 450, 532, and 633 nm light radiation. (**b**) Time-dependent photocurrent measurement on the phototransistors under different power of 633 nm laser, with a bias voltage of 1 V. (**c**) Transient response under the illumination of a 633 nm light. (**d**) Photocurrent and photoresponsivity versus optical illumination power at a wavelength of 633 nm. (**e**) Noise spectral density of the graphene/perovskite phototransitor.

**Figure 6 nanomaterials-11-00641-f006:**
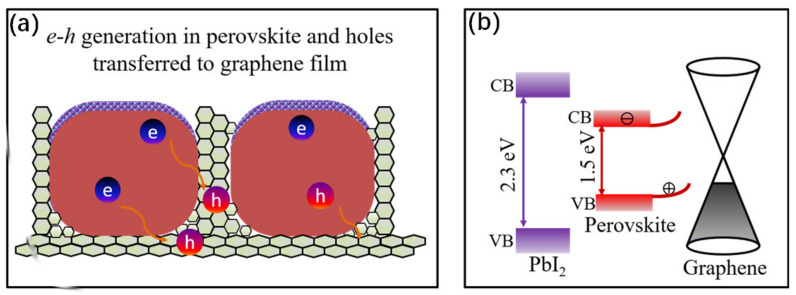
(**a**) Schematic representation and charge transfer process of the device, outside are the passivated PbI2 layer. (**b**) Electronic band diagram of the PbI2/perovskite/graphene nanowalls heterojunction; the injection of the photogenerated holes into the graphene is facilitated by the built-in field formed at the junction.

**Table 1 nanomaterials-11-00641-t001:** Fitting decay times of various perovskite films.

Materials	τ_1_ (ns)	τ_2_ (ns)	Reference
CH_3_NH_3_PbI_3_	4.5 ± 0.3		[38]
CH_3_NH_3_PbI_x_Cl_3-x_		55.3	This work
	4.57		[20]
	3.7~6.1	82.2~101.3	[32]
CH_3_NH_3_PbI_x_Cl_3-x_-Graphene Monolayer	5.0	77.7	This work
	2.11		[38]
CH_3_NH_3_PbI_x_Cl_3-x_-Graphene Nanowalls	3.7	73.7	This work

**Table 2 nanomaterials-11-00641-t002:** Responsivity and time response of perovskite/graphene photodetectors.

Materials	τ_rise_ (ms)	τ_fall_ (ms)	R (A/W)	Detectivity (Jones)	Reference
CH_3_NH_3_PbI_3_ Nanowire	0.1	0.1	4.95 (1 nW/cm^2^@530 nm)	2 × 10^13^	[15]
2D Perovskite	20	40	12 (15.0 mW/cm^2^@532 nm)	Not given	[40]
Graphene/Perovskite	87	540	180 (1 μW@780 nm)	10^9^	[8]
Graphene/Perovskite	~120	~750	6.0 × 10^5^ (1.05 nW@405 nm)	Not given	[20]
CH_3_NH_3_PbI_3_ Nanowire/Graphene	5.5 × 10^4^	7.5 × 10^4^	2.6 × 10^6^ (3.3 pW@633 nm)	Not given	[21]
CH_3_NH_3_PbI_x_Cl_3-x_/Graphene Nanowalls	50	290	2016.7 (40 μW@635 nm)	7.2 × 10^10^	This work

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
