# Peer review of "Lateral Structured Phototransistor Based on Mesoscopic Graphene/Perovskite Heterojunctions"

_nanomaterials, 2021, doi:10.3390/nano11030641_

Round 1

Reviewer 1 Report

The paper "Lateral Structured Phototransistor Based on Mesoscopic Graphene/Perovskite Heterojunctions" by Zhou Dahua et al. is a very careful and interesting study.
The subject is well introducted, the devices fabrication is accurate and the results discussed in the detail.

I would like to add just few comments here:

Line 30-33 - I would suggest mentioning here the 2D/3D graphene/semiconductor heterojunctios, which have been widely proposed and demonstrated as photodetectors and solar cells. See for instance: https://doi.org/10.1016/j.physrep.2015.10.003
and
https://doi.org/10.1088/1361-6463/aac562.

Line 184-187, Authors mention measurements under a solar simulator without any specific of the system, it would be better to give some technical details.

Author Response

Question: Line 30-33 - I would suggest mentioning here the 2D/3D graphene/semiconductor heterojunctios, which have been widely proposed and demonstrated as photodetectors and solar cells. See for instance: https://doi.org/10.1016/j.physrep.2015.10.003 and https://doi.org/10.1088/1361-6463/aac562.

Answer: The 2D/3D graphene/semiconductor heterojunctios have been mentioned at the end of the 1st paragraph, and cited in Ref. 4 and Ref. 6.

Question: Line 184-187, Authors mention measurements under a solar simulator without any specific of the system, it would be better to give some technical details.

Answer: The needed technical details were given in Line 106~109. “To evaluate its photo-detective property, a solar simulator (XES-50S1-EI) and several laser sources (450, 532 and 633 nm) were used as the irradiation light, and a Keithley 2450 and Keithley 4200 was used to record the electrical results, respectively.”

      In addition, more detailed information presented in line184~185, “Under different solar intensity, as 0.5, 0.75 and 1 sun (1000 W/m2), the transistor presented an obvious photo-responsive property.”

Reviewer 2 Report

The manuscript describes perovskite-graphene heterostructures designed as transistor photodetectors. Lateral nanowall graphene structures allow collection of charge, facilitated by the porous structure which allows space for perovskites crystals and a large surface area connected with the perovskite crystalline grain boundaries. The experimental design is described in sufficient detail. However, the meaning is sometimes lost due to incomplete sentences and confusion in the choice of words (responsibility for responsivity, composited for composed, attribute for contribute, cleared for cleaned).

Abstract: “Graphene presented a nanowall structure, and its framework is much appropriated for perovskite crystalline to mount in.” This sentence needs improvement and restructuring.

P1, line 30 and line 31, responsivity.

P2, line 46, is composed layer-by-layer

P2, line 48, can be much more easily fabricated

P2, line 57, photon induced holes are transferred from perovskite to graphene,

P2, line 59, Such effects contribute to

P2, line 67, “The distance between graphene nanowall is comparable with the electron-hole diffusion length in perovskite crystalline, therefore, this device is in favor of carrier collection before the recombination happened”. This sentence needs work, particularly the last part.

P2, line 77, cleaned in acetone

P4, line 141, Please clarify the following: “the perovskite surface presented relatively bright contrast compared the grain center with boundary areas.”

P4, line 146, “less intensive” should be “less intensity”? Also check line 160, “intensity”?

Table 2, please use a consistent symbol for multiplication signs.

P9, line 227, verb needed: “photoexcited holes in perovskites are transferred”. Also line 229, “the holes are injected into the graphene,”. Also line 231, “the decreasing of carrier recombination, hole lifetime is increased, thus enhancing the device’s responsivity.” Line 238, “In the passivation process, PbI2 crystals separate from perovskite.”

P9, line 252, “it achieved faster rise and fall times than”.

Author Response

Question: Abstract: “Graphene presented a nanowall structure, and its framework is much appropriated for perovskite crystalline to mount in.” This sentence needs improvement and restructuring.

Answer: It is improved as: Graphene nanowall shows a porous structure, and the spaces between graphene nanowall is much appropriated for perovskite crystalline to mount in.

Questions:

P1, line 30 and line 31, responsivity.

Answer: P1, line 28, line 29, “responsibility” weree replaced by “responsivity”.

Questions: P2, line 46, is composed layer-by-layer

Answer: P2, line 45, composed layer-by-layer

Questions: P2, line 48, can be much more easily fabricated

Answer: P2, line 47~48, be much mor easily fabricated

Questions: P2, line 57, photon induced holes are transferred from perovskite to graphene,

Answer: P2, line 56, active voice should be used here, as “transfer”

Questions: P2, line 59, Such effects contribute to

Answer: Such effects closely connected with the charge transfer process in the interfacial,

Questions: P2, line 67, “The distance between graphene nanowall is comparable with the electron-hole diffusion length in perovskite crystalline, therefore, this device is in favor of carrier collection before the recombination happened”. This sentence needs work, particularly the last part.

Answer: P2, line 69, new sentence revised, “thus, more excited carriers could be connected by the electrodes but not recombination”

Questions: P2, line 77, cleaned in acetone

Answer: P2, line 77, “cleared” replaced by “cleaned”.

Questions: P4, line 141, Please clarify the following: “the perovskite surface presented relatively bright contrast compared the grain center with boundary areas.”

Answer: P4, line 141~143, “In addition, after the thermal annealing process, for the perovskite grain, one can see that its boundary areas present relatively brighter than its center part.”

Questions: P4, line 146, “less intensive” should be “less intensity”? Also check line 160, “intensity”?

Answer: P4, line 146, and line 160, both “intensive” were replaced by “intensity”

Questions: Table 2, please use a consistent symbol for multiplication signs.

Answer: Multiplication signs are consistent, as “×”

Questions: P9, line 227, verb needed: “photoexcited holes in perovskites are transferred”. Also line 229, “the holes are injected into the graphene,”. Also line 231, “the decreasing of carrier recombination, hole lifetime is increased, thus enhancing the device’s responsivity.” Line 238, “In the passivation process, PbI2 crystals separate from perovskite.”

Answer: We believe “transfer” and “inject”, “separate” should be used as active voice, therefore its corresponding revision were made in line 224, line 228, and line 230, line 238.

Questions: P9, line 252, “it achieved faster rise and fall times than”.

Answer: P9, line 251, it achieved faster rise and fall times than
